# Predictors of singleton preterm birth using multinomial regression models accounting for missing data: A birth registry-based cohort study in northern Tanzania

Innocent B. Mboya[1,2]*, Michael J. Mahande[3], Joseph Obure[3], Henry G. Mwambi[1]

1 School of Mathematics, Statistics and Computer Science, University of KwaZulu-Natal, Pietermaritzburg, South Africa, 2 Department of Epidemiology and Biostatistics, Institute of Public Health, Kilimanjaro Christian Medical University College, Moshi, Tanzania, 3 Department of Obstetrics and Gynecology, Kilimanjaro Christian Medical Center, Moshi, Tanzania

* ib.mboya@gmail.com

## Abstract

### Background

Preterm birth is a significant contributor of under-five and newborn deaths globally. Recent estimates indicated that, Tanzania ranks the tenth country with the highest preterm birth rates in the world, and shares 2.2% of the global proportion of all preterm births. Previous studies applied binary regression models to determine predictors of preterm birth by collapsing gestational age at birth to <37 weeks. For targeted interventions, this study aimed to determine predictors of preterm birth using multinomial regression models accounting for missing data.

### Methods

We carried out a secondary analysis of cohort data from the KCMC zonal referral hospital Medical Birth Registry for 44,117 women who gave birth to singletons between 2000-2015. KCMC is located in the Moshi Municipality, Kilimanjaro region, northern Tanzania. Data analysis was performed using Stata version 15.1. Assuming a nonmonotone pattern of missingness, data were imputed using a fully conditional specification (FCS) technique under the missing at random (MAR) assumption. Multinomial regression models with robust standard errors were used to determine predictors of moderately to late ([32,37) weeks of gestation) and very/extreme (<32 weeks of gestation) preterm birth.

### Results

The overall proportion of preterm births among singleton births was 11.7%. The trends of preterm birth were significantly rising between the years 2000-2015 by 22.2% (95%CI 12.2%, 32.1%, p<0.001) for moderately to late preterm and 4.6% (95%CI 2.2%, 7.0%, p = 0.001) for very/extremely preterm birth category. After imputation of missing values, higher odds of moderately to late preterm delivery were among adolescent mothers (OR = 1.23,

**Data Availability Statement:** The KCMC medical birth registry data cannot be shared publicly because it contains potentially identifying and

sensitive patient information. This has also been stipulated by the local Institutional Review Board of KCMC hospital and the National Ethics Committee in Norway when establishing this birth registry. Permission to use the data in this study was made through the Kilimanjaro Christian Medical University College Research and Ethics Review Committee, and received an approval number 2424. The authors do not have the legal right to share the data publicly. All data requests can be sent to the Executive Director of the KCMC hospital: P. O. Box 3010, Moshi, Tanzania, Email: kcmcadmin@kcmc.ac.tz or through the corresponding author.

**Funding:** This work was funded by GSK Africa Non- Communicable Disease Open Lab through the DELTAS Africa Sub- Saharan African Consortium for Advanced Biostatistics (SSACAB) Grant No. 107754/Z/15/Z- training programme. The views expressed in this publication are those of the author(s) and not necessarily those of GSK. The funders had no role in study design, data collection and analysis, decision to publish, or preparation of the manuscript.

**Competing interests:** The authors have declared that no competing interests exist.

95%CI 1.09, 1.39), with primary education level (OR = 1.28, 95%CI 1.18, 1.39), referred for delivery (OR = 1.19, 95%CI 1.09, 1.29), with pre-eclampsia/eclampsia (OR = 1.77, 95%CI 1.54, 2.02), inadequate (<4) antenatal care (ANC) visits (OR = 2.55, 95%CI 2.37, 2.74), PROM (OR = 1.80, 95%CI 1.50, 2.17), abruption placenta (OR = 2.05, 95%CI 1.32, 3.18), placenta previa (OR = 4.35, 95%CI 2.58, 7.33), delivery through CS (OR = 1.16, 95%CI 1.08, 1.25), delivered LBW baby (OR = 8.08, 95%CI 7.46, 8.76), experienced perinatal death (OR = 2.09, 95%CI 1.83, 2.40), and delivered male children (OR = 1.11, 95%CI 1.04, 1.20). Maternal age, education level, abruption placenta, and CS delivery showed no statistically significant association with very/extremely preterm birth. The effect of (<4) ANC visits, placenta previa, LBW, and perinatal death were more pronounced on the very/extremely preterm compared to the moderately to late preterm birth. Notably, extremely higher odds of very/extreme preterm birth were among the LBW babies (OR = 38.34, 95%CI 31.87, 46.11).

## Conclusions

The trends of preterm birth have increased over time in northern Tanzania. Policy decisions should intensify efforts to improve maternal and child care throughout the course of pregnancy and childbirth towards preterm birth prevention. For a positive pregnancy outcome, interventions to increase uptake and quality of ANC services should also be strengthened in Tanzania at all levels of care, where several interventions can easily be delivered to pregnant women, especially those at high-risk of experiencing adverse pregnancy outcomes.

## Introduction

Every year, an estimated 15 million babies (11%) are born preterm (before 37 completed weeks of gestation) globally [1, 2], majority (81.1%) of these occurs in Asia and sub-Saharan Africa (SSA) [1]. The rates of preterm birth in SSA are notably high in Nigeria (6.9%), Ethiopia (12.0%), and Tanzania (16.6%) [1]. Tanzania ranks the tenth country with the highest preterm birth rates in the world, and shares a 2.2% of the global proportion of all preterm births [1]. The country specific estimates shows that the proportion of preterm birth ranged between 12-13% in Mwanza region [3–6] to as high as 24% among HIV infected women in Dar es Salaam [7].

Preterm birth is a syndrome with a variety of causes, which can be classified into two broad clinical sub-types: spontaneous preterm birth (spontaneous onset of labour or following prelabour premature rupture of membranes) and provider-initiated preterm birth (induction of labor or elective caesarean birth before 37 completed weeks of gestation for maternal or fetal indications, both "urgent" or "discretionary", or other non-medical reasons) [2, 8–11].

A higher risk of preterm birth is reported among women with a history of preterm delivery, those with low (≤24) or high maternal age (≥40), short inter-pregnancy intervals (<24 months), low maternal body mass index (BMI), multiple pregnancies, maternal infections such as urinary tract infections, malaria, bacterial vaginosis, HIV and syphilis and those with inadequate (<4) ANC visits [5, 9, 12–15]. Stress and excessive physical work or long times spent standing, drug abuse such as smoking and excessive alcohol consumption, sex of the child (more among males compared to females), hypertensive disorders of pregnancy such as pre-eclampsia or eclampsia, placental abruption, cholestasis, fetal distress, fetal growth restriction, small for gestational age (a birth weight below the 10th percentile for the gestational age),

and early induction of labor or cesarean birth (before 39 completed weeks of gestation) whether for medical or non-medical reasons also increases the risk of preterm birth [2, 5, 9, 16–18].

Globally, preterm birth is a leading cause of deaths among children under five years of age [1, 2, 10, 19]. SSA is one of the regions with the highest under five deaths in the world [19, 20]. In 2018, preterm birth complications accounted for 18% of death of children under the age of five and 35% of all newborn deaths globally [20]. Preterm birth also increases the risk of babies dying from other causes, especially neonatal infections [9]. Despite modern advances in obstetric and neonatal management, the rate of preterm birth are on the rise in both low-, middle- and high-income countries [1, 2, 21, 22], while in many low- and middle-income countries, preterm newborns are reported to die because of a lack of adequate newborn care [1].

Despite a substantial progress in improving child survival since 1990 [1, 23], preterm birth remains a crucial issue in child mortality and improving quality of maternal and newborn care [1]. To increase child survival and reduce preterm birth complications, the World Health Organization (WHO) recommends essential care during childbirth and postnatal period for every mother and baby (i.e. routine practice for the safe childbirth before, during and after birth), provision of antenatal steroid injections, magnesium sulfate for prevention of cerebral palsy in the infant and child, kangaroo mother care, and antibiotics to treat newborn infections [2, 24]. Tanzania has also adopted these strategies [25, 26] and is one of the five countries where WHO implements a clinical trial on the immediate kangaroo mother care (KMC) for preterm and babies weighing <2000 grams [2, 26].

Epidemiologists are often interested in estimating the risk of adverse events originally measured on an interval scale (such as gestational age in weeks), but they often choose to divide the outcome into two or more categories in order to compute an estimate of effect (risk or odds ratio) [27]. In this study, we applied the multinomial logistic regression models, to show the effect of covariates on several preterm birth categories [2, 22] to avoid the bias that might be introduced by performing a binary analysis. A number of previous studies to assess predictors of preterm birth collapsed all preterm birth categories and performed a binary regression analysis [6, 7, 12, 18, 28–33]. This may introduce potential bias in estimating the effect of covariates on the risk of preterm birth due to a loss of information resulting from collapsing these categories. For a more focused care in the high-risk pregnancies, it is essential to estimate the risk factors for preterm birth, which may differ by the gestational age at birth.

Furthermore, missing data are common in epidemiological and clinical research [34]. Ignoring missing values in the analysis of such data potentially produces biased parameter estimates [34–37]. Stern et. al., [34] further indicated that "missing data in several variables often leads to exclusion of a substantial proportion of the original sample, which in turn causes a substantial loss of precision and power". Therefore, data analysis in this study accounted for missing data, for more precise parameter estimates. The rest of the paper is organized as follows.

## Materials and methods

### Study design, setting and participants

We utilized secondary birth registry data from a prospective cohort of women who delivered singletons in the Kilimanjaro Christian Medical Center (KCMC) between the years 2000-2015. A detailed description of the KCMC Medical birth registry is also available elsewhere [38–43]. Briefly, KCMC is one of the four zonal referral hospitals in the country and is located in the Moshi municipality, Kilimanjaro region, northern Tanzania. The centre primarily receives deliveries of women from the nearby communities, but also referral cases from within and

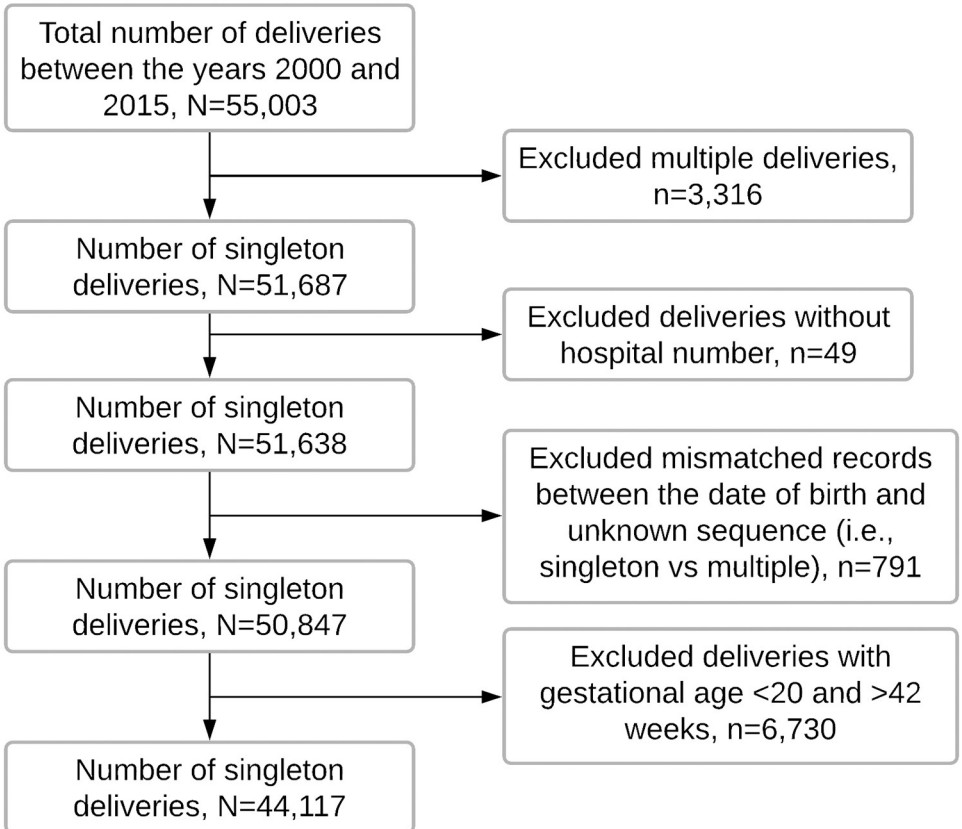

**Fig 1. Flow chart showing the number of deliveries analyzed in this study.** Data from the KCMC Medical Birth registry, 2000-2015.

outside the region. On average, the hospital has approximately 4000 deliveries per year [41, 42, 44].

The study population in this study was singleton deliveries for women of reproductive age (15-49 years) recorded in the KCMC birth registry between 2000-2015, a total of 55,003 deliveries from 43,084 mothers. We excluded 3,316 multiple deliveries, 49 records missing hospital numbers (i.e. unique identification number used to link mothers and their subsequent births), 791 observations with a mismatch between dates of births of children from the same mother or were of unknown sequence (i.e. whether a singleton or multiple births), and 6,730 deliveries with gestational age <20 weeks and >42 weeks. Data was, therefore, analyzed for 44,117 deliveries born from 35,871 mothers (Fig 1).

## Data collection methods

As we have also described the data collection methods elsewhere [43], birth data at KCMC have been recorded using a standardized questionnaire and is collected by specially trained project midwives. The KCMC Medical birth registry collects prospective data for all mothers and their subsequent deliveries in the hospital's department of obstetrics and gynecology. Following informed consent, mothers were interviewed within the first 24 hours after birth given a normal delivery or as soon as a mother has recovered from a complicated delivery. The questionnaire used for data collection is available elsewhere [45]. Although the printed

questionnaires were in the English language, the Project Midwives performing the interviews were well versed in English, Swahili, and one other tribal language. Furthermore, additional information during data collection were extracted from patient files and antenatal cards for more clarification of prenatal information. Data are then transferred, entered and stored in a computerized data base system at the birth registry located at the reproductive health unit of the hospital. A unique identification number was assigned to each woman at first admission and used to trace her medical records at later admissions. Access to data analyzed in this study followed ethical approval granted on June 26, 2019.

## Study variables and variable definitions

The response variable was preterm birth, defined as any birth before 37 completed weeks of gestation and further categorized based on gestational age as <28 weeks (extremely preterm), [28, 32) weeks (very preterm), [32, 37) weeks (moderate to late preterm), and ≥37 weeks (term) for a full-term pregnancy [2]. Gestational age was estimated from the date of last menstrual period of the mother and recorded in completed weeks [4].

Independent variables included maternal background characteristics, particularly age categories (15-19, 20-24, 25-34, 35-39 and 40+) in years, area of residence (rural vs urban), education level (none, primary, secondary and higher), marital status (single, married and widow/divorced), occupation (unemployed, employed and others), parity (primipara vs multipara (para 2-6)), referral status (referred for delivery or not), number of antenatal care visits (<4 and ≥4 visits), and body mass index (underweight [<18.5 Kg/m$^2$], normal weight [18.5–24.9 Kg/m$^2$], overweight [25–29.9 Kg/m$^2$], and obese [≥30 Kg/m$^2$]). Maternal health before and during pregnancy included, alcohol consumption during pregnancy, maternal anemia, malaria, systemic infections/sepsis and pre-eclampsia/eclampsia (all categorized as binary, yes/no). Maternal HIV status was categorized as positive or negative. Complications during pregnancy and delivery included premature rapture of the membranes (PROM), postpartum hemorrhage (PPH), placenta previa and placenta abruption also categorized as binary, yes/no, with "yes" indicating the occurrence of these outcomes. Newborn characteristics included sex (male vs female), perinatal status (dead if experienced stillbirth/early neonatal death vs alive) [43], and low birth weight (LBW) defined as an absolute infant birth weight of <2500g regardless of gestational age at birth [46, 47].

## Statistical and computational analysis

Data were analyzed using STATA version 15.1 (StataCorp LLC, College Station, Texas, USA). The primary unity of analysis was singleton deliveries for women recorded in the KCMC Medical Birth Registry between the years 2000 and 2015. We summarized numeric variables using means and standard deviations, and categorical variables using frequencies and percentages. The Chi-square test was used to compare the proportion of preterm birth by participants characteristics. We used multinomial logistic regression models to determine the predictors of preterm birth as opposed to previous studies [4, 6, 7, 12, 18, 28–30, 32, 33, 48] that performed a binary regression analysis.

The multinomial/polytomous regression model is an extension of the logistic model for binary responses to accommodate multinomial responses which does not have any restrictions on the ordinality of the response [27]. Let $Y_i$ denote a nominal response variable for the $i$th subject, and $Y_i = c$ (the response variable occuring in category $c$), while $Pr(Y_i)$ defines the

probability that $Y_i = c$. The multinomial logit model can be written as

$$P_{ic} = Pr(Y_i = c|X_{ij}) = \frac{\exp{(\eta_{ic})}}{1 + \sum_{c=2}^{C} \exp{(\eta_{ic})}} \quad \text{for} \quad c = 2, 3, \ldots, C \tag{1}$$

$$P_{i1} = Pr(Y_i = 1|X_i) = \frac{1}{1 + \sum_{c=2}^{C} \exp{(\eta_{ic})}} \tag{2}$$

A nominal model to allow for any possible set of $c - 1$ response categories is written as

$$P_{ic} = \frac{\exp{(\eta_{ic})}}{\sum_{c=1}^{C} \exp{(\eta_{ic})}} \quad \text{for} \quad c = 1, 2, \ldots, C \tag{3}$$

where the multinomial logit $\eta_{ic} = X'_{ic}\beta_c$. In this model, all of the effects $\beta_c$ vary across categories ($c = 1, 2, \ldots, C$) and makes comparisons to a reference category compared to the ordinal regression model that uses cumulative comparisons of the categories [49]. We used robust standard errors adjusted for clusters to account for nested observations/ deliveries within mothers.

We would like to indicate here that we performed preliminary analysis using the binary and ordinal logistic regression models. There were a couple of variables that did not satisfy the proportional odds (PO) assumption, hence the ordinal logistic regression model could not be used. The close alternative model that relaxes the PO assumption are the generalized ordered logistic regression models. However, we encountered a non-convergence problem, especially with four preterm birth categories and appropriate interpretation of results. For instance, the order of gestational age categories is <28 weeks (extremely preterm), [28, 32) weeks (very preterm), [32, 37) weeks (moderate to late preterm), and 37+ weeks (term/normal). Assuming the variable is coded as 0 to 3 (with 0 being term birth), the first panel of coefficients will be interpreted as; 0 vs. 1+2+3, then 0+1 vs 2+3 etc [50]. This will imply modeling the probability of delivering at a normal gestational age (category 0) compared to preterm (categories 1-3), probability of delivering term and very preterm vs other preterm categories, etc. Similar interpretations will apply even if preterm birth is coded from extremely preterm (0) to term (3). Such interpretation could be somehow misleading given the nature of this outcome and may not be appealing to clinicians or public health practitioners. Nevertheless, the choice of regression models often depends on the research question one would like to address. In this study, the choice of multinomial regression model was relevant to determine preterm birth predictors across different preterm birth categories, other than performing a binary or an ordinal regression analysis.

As previously indicated, data analysis in this study considered missing values in the covariates. A description of how missing data were imputed is also reported in [43]. Data were imputed using a multiple imputation technique, which is a commonly used method to deal with missing data, which accounts for the uncertainty associated with missing data [34, 37, 51]. We assumed the missing data were missing at random (MAR) where the probability of data being missing does not depend on the unobserved data, conditional on the observed data [34–37]; hence the variables in the dataset were used to predict missingness [43]. We also assumed a nonmonotone pattern of missingness in which some subject values were observed again after a missing value occurs [35, 43, 51]. Under a nonmonotone pattern of missingness, it is recommended to use chained equations, which goes with several names such as the Markov chain Monte Carlo (MCMC), and the fully conditional specification (FCS), to impute missing values [37, 51–55]. Furthermore, the FCS method allows imputation of all types of variables simultaneously, namely some continuous and other categorical.

For the illustration of FCS algorithm, we let $Y$ denote the fully observed outcome in this study i.e., preterm birth, $X$ denote the partially observed covariates $X = X_1, \ldots, X_p$, and $W$ denote the fully observed covariates $W = W_1, \ldots, W_q$. Let $X^o$ and $X^m$ denote the vectors of observed and missing values of $X$ for $n$ subjects. For each partially observed covariate $X_j$, we posit an imputation model $f(X_j|X_{-j}, W, Y, \theta_j)$ with parameter $\theta_j$ where $X_{-j} = (X_1, \ldots, X_{j-1}, X_{j+1}, \ldots, X_p)$ [56]. This according to [56] is typically a generalized linear model chosen according to the type of $X_j$ (e.g. continuous, binary, multinomial, and ordinal). Furthermore, a noninformative prior distribution $f(\theta_j)$ for $\theta_j$ is specified. We further let $x_j^o$ and $x_j^m$ denote the vectors of observed and missing values in $X_j$ for the $n$ subjects and $y$ and $w$ denote the vector and matrix of fully observed values of $Y$ and $W$ across $n$ subjects.

Let $x^{m(t)}$ denote imputations of the missing values $x_j^m$ at iteration $t$ and $x_j^{(t)} = (x_j^o, x_j^{m(t)})$ denote vectors of observed and imputed values at iteration $t$. Let $x_{-j}^{(t)} = (x_1^{(t)}, \ldots, x_{j-1}^{(t)}, x_{j+1}^{(t-1)}, \ldots, x_p^{(t-1)})$. The $t$th iteration of the algorithm consists of drawing from the following distributions (up to constants of proportionality) [56];

$$
\left.
\begin{aligned}
\theta_1^{(t)} &\sim f(\theta_1)f(x_1^o|x_{-1}^{(t)}, w, y, \theta_1) \\
x_1^{m(t)} &\sim f(x_1^m|x_{-1}^{(t)}, w, y, \theta_1^{(t)}) \\
\theta_2^{(t)} &\sim f(\theta_2)f(x_2^o|x_{-2}^{(t)}, w, y, \theta_2) \\
x_2^{m(t)} &\sim f(x_2^m|x_{-2}^{(t)}, w, y, \theta_2^{(t)}) \\
&\vdots \\
\theta_p^{(t)} &\sim f(\theta_p)f(x_p^o|x_{-p}^{(t)}, w, y, \theta_p) \\
x_p^{m(t)} &\sim f(x_p^m|x_{-p}^{(t)}, w, y, \theta_p^{(t)})
\end{aligned}
\right\}
\tag{4}
$$

The FCS starts by calculating the posterior distribution $p(\theta|x^o)$ of $\theta$ given the observed data. This is followed by drawing a value of $\theta^*$ from $p(\theta|x^0)$ given $(x^o, x_{-j}^{(t)}, w, y)$, which is the product of the prior $f(\theta_j)$ and the likelihood corresponding to fitting the imputation model for $X_j$ to subjects for whom $X_j$ is observed, using the observed and most recently imputed values of $X_{-j}$ [56]. Missing values in $X_j$ are then imputed from the imputation model using the parameter value drawn in the preceding step [56]. Finally, a value $x^*$ is drawn from the conditional posterior distribution of $x^m$ given $\theta = \theta^*$. The process is then repeated depending on the desired number of imputations [36, 53, 55, 56]. Within each imputation, there is an iterative estimation process until the distribution of the parameters governing the imputations have converged in the sense of becoming stable, although more cycles may be required depending on certain conditions such as the amount of missing observations in the data [55, 56]. Rubin's rule is then used to provide the final inference for $\hat{\theta}$ by averaging the estimates across $M$ imputations given by [56];

$$
\hat{\theta}_M = \frac{\sum_{m=1}^{M} \hat{\theta}^m}{M}
\tag{5}
$$

while the estimate of the variance of $\hat{\theta}^M$ is given by;

$$
\widehat{Var}(\hat{\theta}_M) = \left[\frac{1}{M}\sum_{m=1}^{M}\widehat{Var}(\hat{\theta}^m)\right] + \left[(1+1/M)\frac{1}{M-1}\sum_{m=1}^{M}(\hat{\theta}^m - \hat{\theta}_M)^2\right]
\tag{6}
$$

which is a combination of within and between imputation variances. Detailed descriptions on

implementation of the FCS/MICE algorithm in STATA is well-presented elsewhere [54, 57]. Maternal age and education level were imputed as ordinal variables, while maternal occupation, marital status, and BMI (because normal weight (18.5–24.9 Kg/m$^2$) was a reference category) as multinomial variable [43]. The rest of the variables were binary, and so imputed using the binomial distribution. Preterm birth (the outcome in this study), parity, pre-eclampsia/eclampsia, anemia, malaria, systemic infections/sepsis, PROM, PPH, abruption placenta, placenta previa, and year of birth did not contain any missing values, hence used as auxiliary variables in the imputation model. The imputation model generated 20 imputed datasets after 500 iterations (imputation cycles). A random seed of 5000 was specified for replication of imputation results each time a multiple imputation analysis is performed [51].

We developed a multivariable analysis model by including all covariates in the multinomial logit analysis model [54]), with standard errors adjusted for clusters (i.e., deliveries nested within mothers). We then performed stepwise regression, in which variables with $p < 0.1$ or $p < 10\%$ were retained in the model. The next steps entailed performing a series of adjusted analysis to test the effect of retaining and dropping variables in the multivariable model. Variables in the final model were evaluated at p-value<0.05 level of statistical significance. We used AIC to compare model performance and non-nested models [58], and Likelihood ratio test to compare nested models. After the imputation of missing values, we estimated parameter estimates adjusting for the variability between imputations [54, 57]. Before the analysis of imputed data, we firstly performed complete case analysis using multivariable multinomial regression model. The final model from this analysis was then compared to those from the multiply imputed dataset. We followed the recommendations suggested by Sterne et al., [34] for reporting and analysis of missing data.

## Ethical consideration

As described in [43], this study was approved by the Kilimanjaro Christian Medical University College Research Ethics and Review Committee (KCMU-CRERC) with approval number 2424. For practical reasons, since the interview was administered just after the woman had given birth, consent was given orally. The midwife-nurse gave every woman oral information about the birth registry, the data needed to be collected from them, and the use of the data for research purposes. Women were also informed about the intention to gather new knowledge, which will, in turn, benefit mothers and children in the future. Participation was voluntary and had no implications on the care women would receive. Following consent, mothers were free to refuse to reply to single questions. For privacy and confidentiality, unique identification numbers were used to both identity and then link mothers with child records. There was no any person-identifiable information in any electronic database, and instead, unique identification numbers were used. Necessary measures were taken by midwives to ensure privacy during the interview process.

## Results

### Maternal background characteristics by gestational age categories

The overall proportion of preterm birth in this study was 12.8%, of which 9.8% children were born at [32, 37) weeks (moderate to late preterm), 1.6% at [28, 32) weeks (very preterm), and 0.4% at <28 weeks (extremely preterm) of gestation. The proportions of preterm birth differed significantly by maternal background and obstetric care characteristics (Tables 1 and 2, respectively). Among adolescent mothers (15-19 years), 12.3% delivered at [32, 37) weeks and 1.8% at [28, 32) weeks of gestation, which is almost similar to that among older mothers (40+ years). The proportion of women who delivered at [32, 37) weeks of gestation was 10.8% among rural

Table 1. Maternal background characteristics by gestational age categories (N = 44,117).

| Characteristics | Total (%) | Gestational age at birth | | | | p-value |
|---|---|---|---|---|---|---|
| | | $\geq$37 | 32-<37 | 28-<32 | <28 | |
| **Mother's age groups (years)** * | | | | | | <0.001 |
| 15-19 | 3637 (8.3) | 3101 (85.3) | 447 (12.3) | 67 (1.8) | 22 (0.6) | |
| 20-24 | 11113 (25.2) | 9797 (88.2) | 1108 (10.0) | 171 (1.5) | 37 (0.3) | |
| 25-34 | 22767 (51.7) | 20321 (89.3) | 2031 (8.9) | 342 (1.5) | 73 (0.3) | |
| 35-39 | 5262 (12.0) | 4576 (87.0) | 556 (10.6) | 110 (2.1) | 20 (0.4) | |
| 40+ | 1267 (2.9) | 1080 (85.2) | 158 (12.5) | 21 (1.7) | 8 (0.6) | |
| **Current area of residence** * | | | | | | <0.001 |
| Rural | 18083 (41.1) | 15690 (86.8) | 1951 (10.8) | 360 (2.0) | 82 (0.5) | |
| Urban | 25935 (58.9) | 23155 (89.3) | 2349 (9.1) | 352 (1.4) | 79 (0.3) | |
| **Mother's highest education level** * | | | | | | <0.001 |
| None | 640 (1.5) | 544 (85.0) | 74 (11.6) | 19 (3.0) | 3 (0.5) | |
| Primary | 24038 (54.6) | 20857 (86.8) | 2654 (11.0) | 426 (1.8) | 101 (0.4) | |
| Secondary | 5406 (12.3) | 4752 (87.9) | 540 (10.0) | 102 (1.9) | 12 (0.2) | |
| Higher | 13967 (31.7) | 12730 (91.1) | 1028 (7.4) | 164 (1.2) | 45 (0.3) | |
| **Occupation** * | | | | | | 0.04 |
| Unemployed | 9617 (21.9) | 8397 (87.3) | 1020 (10.6) | 161 (1.7) | 39 (0.4) | |
| Employed | 31233 (71.2) | 27618 (88.4) | 2999 (9.6) | 502 (1.6) | 114 (0.4) | |
| Others | 3023 (6.9) | 2701 (89.3) | 269 (8.9) | 45 (1.5) | 8 (0.3) | |
| **Marital Status** * | | | | | | <0.001 |
| Single | 5202 (11.8) | 4490 (86.3) | 572 (11.0) | 112 (2.2) | 28 (0.5) | |
| Married | 38697 (88.0) | 34279 (88.6) | 3698 (9.6) | 589 (1.5) | 131 (0.3) | |
| Widowed/Divorced | 87 (0.2) | 62 (71.3) | 18 (20.7) | 5 (5.7) | 2 (2.3) | |
| **Body mass index categories** * | | | | | | <0.001 |
| Underweight (<18.5) | 1582 (5.2) | 1382 (87.4) | 167 (10.6) | 30 (1.9) | 3 (0.2) | |
| Normal weight (18.5-24.9) | 16417 (53.9) | 14735 (89.8) | 1439 (8.8) | 201 (1.2) | 42 (0.3) | |
| Overweight (25-29.9) | 8510 (27.9) | 7763 (91.2) | 633 (7.4) | 94 (1.1) | 20 (0.2) | |
| Obese ($\geq$30) | 3947 (13.0) | 3581 (90.7) | 307 (7.8) | 48 (1.2) | 11 (0.3) | |
| **Total (row %)** | | 38933 (88.2%) | 4309 (9.8%) | 714 (1.6%) | 161 (0.4%) | |

* Variables with missing values.

residents, 11.0% among those with primary education level, 9.6% among those employed, and 9.6% among mothers who were married (Table 1).

## Diseases and complications during pregnancy and delivery by gestational age categories

The diseases and complications during pregnancy and delivery by gestational age categories are shown in (Table 2). There were statistically significant differences in the proportion of preterm birth categories by diseases and complications during pregnancy and delivery except for anaemia, infections/ sepsis and child's sex. Significantly higher proportion of deliveries born at [32, 37) weeks of gestation was among mothers who experienced placenta previa (39.6%), abruption placenta (37.3%), delivered LBW baby (37.1%), perinatal death (28.1%), pre-eclampsia/eclampsia mothers (24.3%), PROM (18.9%) with <4 ANC visits (17.0%), and post-partum hemorrhage (14.8%). Also, the proportion of deliveries born at [28, 32) weeks of gestation was significantly higher among mothers with pre-eclampsia/eclampsia (6.2%), abruption

**Table 2. Diseases and complications during pregnancy and delivery by gestational age categories (N = 44,117).**

| Characteristics | Total (%) | Gestational age at birth | | | | p-value |
|---|---|---|---|---|---|---|
| | | ≥37 | 32-<37 | 28-<32 | <28 | |
| **Pre-eclampsia/eclampsia** | | | | | | <0.001 |
| No | 42282 (95.8) | 37674 (89.1) | 3864 (9.1) | 600 (1.4) | 144 (0.3) | |
| Yes | 1835 (4.2) | 1259 (68.6) | 445 (24.3) | 114 (6.2) | 17 (0.9) | |
| **Anaemia** | | | | | | 0.63 |
| No | 43427 (98.4) | 38331 (88.3) | 4238 (9.8) | 699 (1.6) | 159 (0.4) | |
| Yes | 690 (1.6) | 602 (87.2) | 71 (10.3) | 15 (2.2) | 2 (0.3) | |
| **Malaria** | | | | | | 0.002 |
| No | 38145 (86.5) | 33579 (88.0) | 3785 (9.9) | 637 (1.7) | 144 (0.4) | |
| Yes | 5972 (13.5) | 5354 (89.7) | 524 (8.8) | 77 (1.3) | 17 (0.3) | |
| **Infections** | | | | | | 0.37 |
| No | 43352 (98.3) | 38244 (88.2) | 4243 (9.8) | 706 (1.6) | 159 (0.4) | |
| Yes | 765 (1.7) | 689 (90.1) | 66 (8.6) | 8 (1.0) | 2 (0.3) | |
| **HIV Status** * | | | | | | 0.003 |
| Negative | 32000 (94.8) | 28367 (88.6) | 3047 (9.5) | 472 (1.5) | 114 (0.4) | |
| Positive | 1769 (5.2) | 1521 (86.0) | 213 (12.0) | 31 (1.8) | 4 (0.2) | |
| **Consumed alcohol during pregnancy** * | | | | | | <0.001 |
| No | 31287 (71.8) | 27472 (87.8) | 3150 (10.1) | 543 (1.7) | 122 (0.4) | |
| Yes | 12292 (28.2) | 10998 (89.5) | 1099 (8.9) | 158 (1.3) | 37 (0.3) | |
| **Number of ANC visits** * | | | | | | <0.001 |
| ≥4 | 29490 (68.0) | 27489 (93.2) | 1830 (6.2) | 125 (0.4) | 46 (0.2) | |
| <4 | 13884 (32.0) | 10879 (78.4) | 2366 (17.0) | 540 (3.9) | 99 (0.7) | |
| **Parity** | | | | | | 0.001 |
| Primipara | 35871 (81.3) | 31599 (88.1) | 3519 (9.8) | 606 (1.7) | 147 (0.4) | |
| Multipara | 8246 (18.7) | 7334 (88.9) | 790 (9.6) | 108 (1.3) | 14 (0.2) | |
| **PROM** | | | | | | <0.001 |
| No | 43157 (97.8) | 38187 (88.5) | 4128 (9.6) | 681 (1.6) | 161 (0.4) | |
| Yes | 960 (2.2) | 746 (77.7) | 181 (18.9) | 33 (3.4) | 0 (0.0) | |
| **PPH** | | | | | | <0.001 |
| No | 43874 (99.4) | 38739 (88.3) | 4273 (9.7) | 702 (1.6) | 160 (0.4) | |
| Yes | 243 (0.6) | 194 (79.8) | 36 (14.8) | 12 (4.9) | 1 (0.4) | |
| **Abruption placenta** | | | | | | <0.001 |
| No | 43967 (99.7) | 38857 (88.4) | 4253 (9.7) | 699 (1.6) | 158 (0.4) | |
| Yes | 150 (0.3) | 76 (50.7) | 56 (37.3) | 15 (10.0) | 3 (2.0) | |
| **Placenta previa** | | | | | | <0.001 |
| No | 44021 (99.8) | 38891 (88.3) | 4271 (9.7) | 698 (1.6) | 161 (0.4) | |
| Yes | 96 (0.2) | 42 (43.8) | 38 (39.6) | 16 (16.7) | 0 (0.0) | |
| **Perinatal status** * | | | | | | <0.001 |
| Alive | 42230 (96.0) | 37868 (89.7) | 3796 (9.0) | 462 (1.1) | 104 (0.2) | |
| Died | 1780 (4.0) | 975 (54.8) | 500 (28.1) | 250 (14.0) | 55 (3.1) | |
| **Birth weight** * | | | | | | <0.001 |
| NBW | 39202 (89.1) | 36543 (93.2) | 2500 (6.4) | 107 (0.3) | 52 (0.1) | |
| LBW | 4801 (10.9) | 2334 (48.6) | 1779 (37.1) | 585 (12.2) | 103 (2.1) | |
| **Sex of the baby** * | | | | | | 0.48 |
| Male | 22684 (51.6) | 20032 (88.3) | 2216 (9.8) | 349 (1.5) | 87 (0.4) | |
| Female | 21242 (48.4) | 18743 (88.2) | 2070 (9.7) | 359 (1.7) | 70 (0.3) | |
| **Refereed for delivery** * | | | | | | <0.001 |

(*Continued*)

**Table 2.** (Continued)

| Characteristics | Total (%) | Gestational age at birth | | | | p-value |
| --- | --- | --- | --- | --- | --- | --- |
| | | ≥37 | 32-<37 | 28-<32 | <28 | |
| Yes | 9610 (22.6) | 7883 (82.0) | 1382 (14.4) | 278 (2.9) | 67 (0.7) | |
| No | 32878 (77.4) | 29575 (90.0) | 2807 (8.5) | 409 (1.2) | 87 (0.3) | |
| **Total** | | **38933 (88.2%)** | **4309 (9.8%)** | **714 (1.6%)** | **161 (0.4%)** | |

* Variables with missing values.

placenta (10.0%), placenta previa (16.7%), experienced perinatal death (14.0%), and those who delivered a LBW baby (12.2%).

## Distribution of missing values

Percentage distribution of missing values in this study are summarized in Table 3. Maternal BMI (31.0%) and HIV status (23.5%) accounted for more than half (54.5%) of all missing values. The proportion of missing values was 3.7%, 1.7% and 1.2% for referral status, number of ANC visits and alcohol consumption during pregnancy, respectively. The rest of the variables had less than 1% of missing values.

## Trends of preterm birth from 2000–2015

The proportion of moderate to late preterm (32 to <37) and very preterm (28 to <32) increased significantly over the years between 2000-2015 (Fig 2). The annual increase of PTB at [32,37) weeks of gestation was 22.2% (95%CI 12.2%, 32.1%, p<0.001) while for [28,32) weeks of gestation was 4.2% (95%CI 1.9%, 6.6%, p = 0.002). Despite a slight increasing trend of extremely preterm birth (<28 weeks) deliveries, this increase was not statistically significant (p = 0.37).

However, further analysis of the trends in the very/ extremely preterm birth (i.e., all deliveries at <32 weeks of gestation) revealed a significant annual increase of 4.6% (95%CI 2.2%, 7.0%, p = 0.001). Regression analysis both before and after imputation of missing values, considered two preterm birth categories, i.e., <32 weeks (combined <28 and [28,32) weeks and

**Table 3. Distribution of missing values, KCMC medical birth registry, 2000–2015 (N = 44,117).**

| Variable | Frequency | Percent Missing |
| --- | --- | --- |
| Body Mass Index (BMI) | 13,661 | 31.0 |
| HIV status | 10,348 | 23.5 |
| Referral status | 1,629 | 3.7 |
| Number of antenatal care visits | 743 | 1.7 |
| Consumed alcohol during pregnancy | 538 | 1.2 |
| Occupation | 244 | 0.6 |
| Sex of the child | 191 | 0.4 |
| Marital status | 131 | 0.3 |
| Birth weight of the child | 114 | 0.3 |
| Area of residence | 107 | 0.2 |
| Perinatal status | 99 | 0.2 |
| Maternal education level | 71 | 0.2 |
| Maternal age categories | 66 | 0.2 |

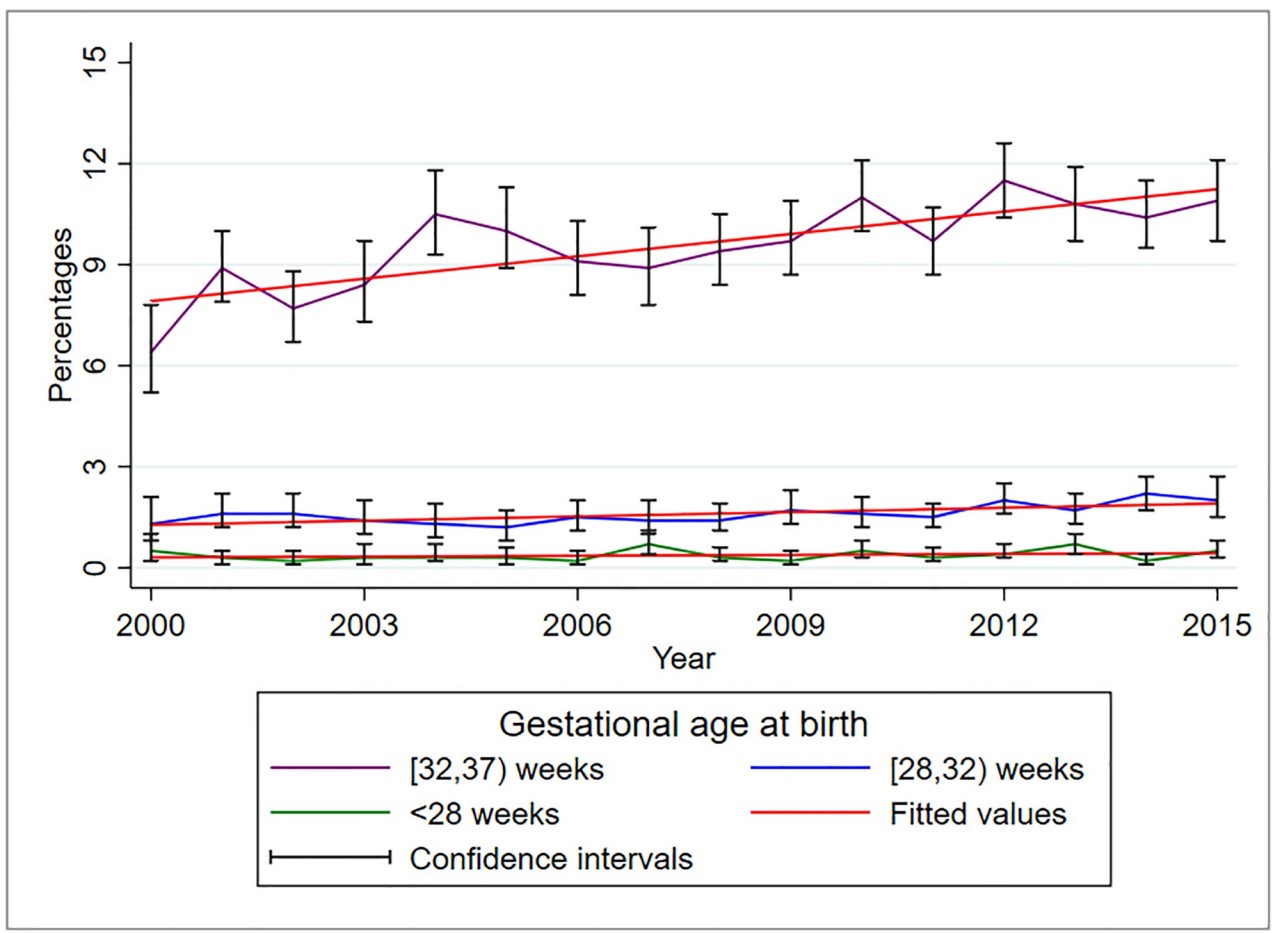

**Fig 2. Trends of preterm birth from 2000–2015 in the KCMC medical birth registry (N = 44,117).**

[32,37] weeks) compared to ≥37 weeks, due to small sample size in the <28 category and increased statistical power to detect the observed effect.

## Predictors of preterm birth

Due to a small number of deliveries 161 (0.4%) at <28 weeks of gestation recorded at the KCMC Medical birth registry between 2000 and 2015, we combined this category with deliveries at [28,32) weeks of gestation, 714 (1.6%). This gives a total of 875 (2.0%) in the new <32 (very/extremely preterm) category. The collapsed categories increased statistical power and improved model performance, given a non-convergence problem of models with all three preterm birth categories.

**Results before imputation of missing values.** Findings from the adjusted analysis of the multinomial regression model before imputation of missing values are shown in Table 4. The standard errors are robust (adjusted) to clustering of deliveries within mothers. Higher odds of delivering at [32,37) weeks of gestation (moderate to late preterm) were among adolescent (15-19) mothers (OR = 1.29, 95% 1.13, 1.48) and those aged 20-24 years (OR = 1.17, 95%CI 1.07, 1.28) compared to those aged 25-34 years and those with primary education level (OR = 1.28, 95%CI 1.17, 1.39) compared to higher education level. Also, mothers referred for delivery (OR = 1.20, 95%CI 1.10, 1.31), with pre-eclampsia/eclampsia (OR = 1.88, 95%CI 1.63,

**Table 4. Adjusted analysis for predictors of preterm birth using multinomial regression model before imputation of missing values (N = 41,271).**

| Characteristics | 32-<37 vs. ≥37 weeks | | <32 vs. ≥37 weeks | |
|---|---|---|---|---|
| | AOR† (SE‡) | 95%CI | AOR† (SE‡) | 95%CI |
| **Mother's age groups (years)** | | | | |
| 15-19 | 1.29 (0.09) | 1.13,1.48*** | 1.37 (0.20) | 1.03,1.81* |
| 20-24 | 1.17 (0.05) | 1.07,1.28*** | 1.11 (0.12) | 0.91,1.37 |
| 25-34 | 1 | | 1 | |
| 35-39 | 1.03 (0.06) | 0.92,1.15 | 1.04 (0.13) | 0.82,1.33 |
| 40+ | 1.13 (0.11) | 0.93,1.38 | 0.84 (0.20) | 0.53,1.33 |
| **Maternal highest education level** | | | | |
| None | 1.15 (0.17) | 0.85,1.54 | 1.43 (0.37) | 0.86,2.38 |
| Primary | 1.28 (0.06) | 1.17,1.39*** | 1.11 (0.11) | 0.91,1.35 |
| Secondary | 1.11 (0.07) | 0.98,1.26 | 0.97 (0.14) | 0.74,1.28 |
| Higher | 1 | | 1 | |
| **Referred for delivery (Yes)** | 1.20 (0.05) | 1.10,1.31*** | 1.30 (0.12) | 1.08,1.55** |
| **Pre-eclampsia/eclampsia (Yes)** | 1.88 (0.13) | 1.63,2.15*** | 1.51 (0.19) | 1.18,1.92*** |
| **Number of ANC visits (<4)** | 2.56 (0.10) | 2.38,2.75*** | 5.55 (0.53) | 4.61,6.69*** |
| **Parity (Primipara)** | 0.89 (0.04) | 0.80,0.98* | 0.96 (0.11) | 0.76,1.21 |
| **PROM (Yes)** | 1.83 (0.18) | 1.51,2.22*** | 1.51 (0.35) | 0.96,2.39 |
| **Abruption placenta (Yes)** | 2.01 (0.49) | 1.24,3.24** | 1.60 (0.57) | 0.80,3.20 |
| **Placenta previa (Yes)** | 4.90 (1.46) | 2.73,8.77*** | 8.68 (3.72) | 3.75,20.10*** |
| **Delivery mode (CS)** | 1.16 (0.04) | 1.07,1.25*** | 0.93 (0.08) | 0.78,1.11 |
| **Birth weight (LBW)** | 8.05 (0.34) | 7.41,8.75*** | 36.23 (3.55) | 29.91,43.89*** |
| **Sex of the baby (Male)** | 1.11 (0.04) | 1.03,1.19** | 1.22 (0.10) | 1.04,1.43* |
| **Perinatal death (Yes)** | 2.06 (0.15) | 1.78,2.37*** | 5.38 (0.55) | 4.41,6.56*** |
| **Year** | 1.02 (0.00) | 1.01,1.03*** | 1.06 (0.01) | 1.04,1.09*** |

\* $p < 0.05$,

\*\* $p < 0.01$,

\*\*\* $p < 0.001$.

†AOR: Adjusted Odds Ratio, adjusted for maternal age groups (years), highest level of education, referral status, pre-eclampsia/eclampsia, number of ANC visits, parity, PROM, abruption placenta, placenta previa, delivery mode, child's birth weight, perinatal status and year of birth.

‡SE: Standard errors adjusted for clustering of deliveries within mothers.

2.15), with inadequate (<4) ANC visits (OR = 2.56, 95%CI 2.38, 2.75), experienced PROM (OR = 1.83, 95%CI 1.51, 2.22), abruption placenta (OR = 2.01, 95%CI 1.24 3.24), placenta previa (OR = 4.90, 95%CI 2.73, 8.77), delivered through cesarean section (OR = 1.16, 95%CI 1.07, 1.25), delivered a LBW baby (OR = 8.05, 95%CI 7.41, 8.75), experienced perinatal death (OR = 2.06, 95%CI 1.78, 2.37), and delivered a male child (OR = 1.11, 95%CI 1.03, 1.19), compared to their respective reference levels had higher odds of delivering moderate to late preterm birth. Primiparous women were less likely to deliver moderate to late preterm (OR = 0.89, 95%CI 0.80, 0.98). For every year increase, the odds of delivering at 32-<37 weeks of gestation increased significantly by 2% (OR = 1.02, 95%CI 1.01, 1.03).

Moreover, in the adjusted analysis, maternal age, referral status, pre-eclampsia/eclampsia, number of ANC visits, placenta previa, LBW, perinatal status, child's sex, and year of birth remained significantly associated with delivering at <32 weeks of gestation (very/extremely preterm). Notably, the odds of delivering at <32 of gestation were nearly forty times (OR = 36.23, 95%CI 29.91, 43.89) among deliveries born with LBW compared to normal weight at birth. This is more than four times higher odds compared to the effect in the

gestational age of [32,37) weeks. Mothers aged 15-19 years (OR = 1.37, 95%CI 1.03, 1.81), referred for delivery (OR = 1.30, 95%CI 1.08, 1.55), with pre-eclampsia/eclampsia (OR = 1.51, 95%CI 1.18, 1.92), with inadequate (<4) ANC visits (OR = 5.55, 95%CI 4.61, 6.69), experienced placenta previa (OR = 8.68, 95%CI 3.75, 20.10), experienced perinatal death (OR = 5.38, 95%CI 4.41, 6.56), and delivered male children (OR = 1.22, 95%CI 1.04, 1.43) had higher odds of delivering very/extremely preterm birth (<32 weeks of gestation) as compared to their counterparts. Furthermore, for every year increase, the odds of delivering at <32 weeks of gestation increased significantly by 6% (OR = 1.06, 95%CI 1.04, 1.09), which is three-times higher than the effect in the [32,37) weeks of gestation. These results demonstrate the advantage of the multinomial regression as opposed to the simple binary regression models. We see that the effect of some covariates (LBW, inadequate ANC visits, placenta previa, and perinatal death) are more pronounced for the extreme preterm birth category than the moderately to late preterm birth category (Table 4).

**Results after imputation of missing values.** After imputation of missing values (in the covariates), the standard errors were relatively lower while the coefficients (odds ratios) (Table 5) were either lower or higher compared to those in the complete case analysis

**Table 5. Adjusted analysis for predictors of preterm birth using multinomial regression model after imputation of missing values (N = 42,089).**

| Characteristics | 32-<37 vs. ≥37 weeks | | <32 vs. ≥37 weeks | |
|---|---|---|---|---|
| | AOR[†] (SE[‡]) | 95%CI | AOR[†] (SE[‡]) | 95%CI |
| **Mother's age groups (years)** | | | | |
| 15-19 | 1.29 (0.09) | 1.13,1.47*** | 1.30 (0.18) | 0.99,1.71 |
| 20-24 | 1.15 (0.05) | 1.06,1.26** | 1.13 (0.11) | 0.93,1.38 |
| 25-34 | 1 | | 1 | |
| 35-39 | 1.03 (0.06) | 0.92,1.15 | 1.06 (0.13) | 0.84,1.34 |
| 40+ | 1.11 (0.11) | 0.91,1.34 | 0.92 (0.20) | 0.60,1.40 |
| **Maternal highest education level** | | | | |
| None | 1.11 (0.17) | 0.82,1.49 | 1.44 (0.36) | 0.88,2.34 |
| Primary | 1.27 (0.06) | 1.17,1.39*** | 1.09 (0.10) | 0.91,1.32 |
| Secondary | 1.10 (0.07) | 0.98,1.25 | 1.00 (0.14) | 0.77,1.32 |
| Higher | 1 | | 1 | |
| **Referred for delivery (Yes)** | 1.20 (0.05) | 1.10,1.30*** | 1.28 (0.11) | 1.08,1.52** |
| **Pre-eclampsia/eclampsia (Yes)** | 1.86 (0.13) | 1.62,2.13*** | 1.61 (0.19) | 1.27,2.03*** |
| **Number of ANC visits (<4)** | 2.56 (0.10) | 2.38,2.75*** | 5.64 (0.54) | 4.67,6.80*** |
| **Parity (Primipara)** | 0.90 (0.04) | 0.82,0.99* | 0.98 (0.11) | 0.78,1.23 |
| **PROM (Yes)** | 1.87 (0.18) | 1.55,2.26*** | 1.63 (0.36) | 1.06,2.50* |
| **Abruption placenta (Yes)** | 1.98 (0.48) | 1.23,3.19** | 1.46 (0.52) | 0.73,2.93 |
| **Placenta previa (Yes)** | 4.76 (1.35) | 2.73,8.28*** | 8.07 (3.32) | 3.61,18.07*** |
| **Delivery mode (CS)** | 1.16 (0.04) | 1.08,1.25*** | 0.91 (0.08) | 0.77,1.08 |
| **Birth weight (LBW)** | 8.09 (0.34) | 7.45,8.78*** | 38.21 (3.67) | 31.65,46.14*** |
| **Perinatal death (Yes)** | 2.10 (0.15) | 1.83,2.42*** | 5.29 (0.52) | 4.37,6.40*** |
| **Sex of the baby (Male)** | 1.11 (0.04) | 1.04,1.20** | 1.22 (0.10) | 1.05,1.43* |
| **Year** | 1.02 (0.00) | 1.01,1.03*** | 1.04 (0.01) | 1.02,1.06*** |

* p<0.05,

** p<0.01,

*** p<0.001.

[†]AOR: Adjusted Odds Ratio, adjusted for maternal age groups (years), highest level of education, referral status, pre-eclampsia/eclampsia, number of ANC visits, PROM, abruption placenta, placenta previa, delivery mode, child's birth weight, perinatal status and year of birth.

[‡]SE: Standard errors adjusted for clustering of deliveries within mothers.

(Table 4). Results from the imputed data indicated significantly higher odds of moderately to late preterm delivery (32 to <37 weeks) were among adolescent mothers aged 15-19 years (OR = 1.29, 95%CI 1.13, 1.3479), aged 20-24 years (OR = 1.15, 95%CI1.06, 1.26), with primary education level (OR = 1.27, 95%CI 1.17, 1.39), and referred for delivery (OR = 1.20, 95%CI 1.10, 1.30). Also, significantly higher odds of moderately to late preterm delivery were among mothers with pre-eclampsia/eclampsia (OR = 1.86, 95%CI 1.62, 2.13), inadequate (<4) ANC visits (OR = 2.56, 95%CI 2.38, 2.75), experienced PROM (OR = 1.87, 95%CI 1.55, 2.26), abruption placenta (OR = 1.98, 95%CI 1.23, 3.19), and placenta previa (OR = 4.76, 95%CI 2.73, 8.28). Likewise, delivery through CS (OR = 1.16, 95%CI 1.08, 1.25), delivering LBW baby (OR = 8.09, 95%CI 7.45, 8.78), experiencing perinatal death (OR = 2.10, 95%CI 1.83, 2.42), and delivering male children (OR = 1.11, 95%CI 1.04, 1.20) were associated with higher odds of delivering moderately to late preterm. Primiparous women were less likely to deliver moderately to late preterm (OR = 0.90, 95%CI 0.82, 0.99) compared to multiparous. For every one year increase, the odds of delivering moderately to late preterm increased significantly by 2% (OR = 1.02, 95%CI 1.01, 1.03), Table 5.

Furthermore, after imputation of missing values the positive effect of PROM on very/ extremely preterm birth (<32 weeks of gestation) is observed to be statistically significant (OR = 1.63, 95%CI 1.06, 2.50) compared to results before imputation of missing values (Table 4). Significantly higher odds of very/extreme preterm birth was among mothers referred for delivery (OR = 1.28, 95%CI 1.08, 1.52), with pre-eclampsia/eclampsia (OR = 1.61, 95%CI 1.27, 1.03), inadequate (<4) ANC visits (OR = 5.64, 95%CI 4.67, 6.80), experienced placenta previa (OR = 8.07, 95%CI 3.61, 18.07), delivered LBW baby (OR = 38.21, 95%CI 31.65, 46.14), experienced perinatal death (OR = 5.29, 95%CI 4.37, 6.40), and delivered male children (OR = 1.22, 95%CI 1.05, 1.43). Also, the odds of delivering very/ extreme preterm birth increased significantly by 4% (OR = 1.04, 95%CI 1.02, 1.06) for every calendar year.

## Discussion

Globally, the trends of preterm birth rate has been increasing over time [1, 2, 9, 48]. Findings in the current study also revealed the rising trends of both moderate to late preterm (32 to <37 weeks of gestation) and very/extremely preterm birth (<32 weeks of gestation) between the years 2000-2015. A recent systematic review and modelling analysis revealed that Tanzania is among the top 10 countries (10th position) with the highest preterm birth rate (16·6%) and contributed to 2.2% of the global preterm birth estimates [1]. Based on the estimates released seven years ago (2013) by Blencowe et. al., [9], Tanzania was not in the top 10 countries with the highest (>15%) preterm birth rates globally. By then, Malawi had the highest preterm birth rate (18%) in SSA and South East Asia [9, 12].

Previous studies at the KCMC zonal referral hospital [4, 5] and Bugando Medical Center in Mwanza region [6] reported the preterm birth rate of 14%; where [4] utilized cohort data between the years 2000-2008 while [5] and [6] conducted case-control studies. The rising trends and relatively high preterm birth rates in Tanzania are alarming, given the documented short- and long-term consequences, particularly an increased risk of recurrence in subsequent pregnancies, stillbirths, and neonatal mortality [4, 11, 12, 19, 32, 59, 60]. In fact, mothers who experienced perinatal death in this study were more likely to deliver preterm. The effect of perinatal death almost doubled in the very/ extremely preterm category.

Multiple imputation was performed to increase precision of parameter estimates, as it accounts for the uncertainty associated with missing data [34, 35, 37, 51]. After the imputation of missing values, the standard errors are relatively lower and coefficients (odds ratios) were either lower or higher than those in the complete case analysis. Although the direction of

associations remained the same, precision of parameters estimates is increased after imputation of missing data. It has been reported that "multiple imputation provides unbiased and valid estimates of associations based on information from the available data—ie, yielding estimates similar to those calculated from full data" [37]. Data analysts should consider accounting for missing data in their analysis using proper techniques to reduce the bias associated with simple analysis (such as analyzing available or complete cases) that ignore missing values [37, 51, 52].

Results from the imputed data revealed that adolescent (15-19 years) mothers and mothers aged 20-24 years had higher odds of delivering moderately to late preterm births (32 to <37 weeks) as well as very/extremely preterm (<32 weeks though this association was not statistically significant) compared to mothers aged 25-34 years. Our findings are consistent with previous studies [7, 12, 15, 29, 48]. Authors in these studies revealed that younger (<24 years) mothers are at increased risk of delivering preterm. A previous study in Canada indicated that women aged 20-24 years were more at risk of delivering spontaneous preterm birth [15]. However, authors in this study did not include adolescent mothers. Data from the Tanzania Demographic and Health Survey 2015/16 revealed the rising trends of teenage childbearing (15-19 years) from 23% in 2010 to 27% in 2015/16 [61]. Younger age at first pregnancy is a public health concern due to an increased risk of complications during pregnancy and child birth as well as maternal and neonatal mortality [15, 61]. A systematic review and meta-analysis in SSA documented an association between adolescent child-bearing and an increased risk of low birth weight, pre-eclampsia/eclampsia, preterm birth and maternal and perinatal mortality [62]. Our findings suggests that interventions in Tanzania should emphasize on delayed age at first pregnancy and provision of adolescent and youth friendly sexual and reproductive health services [26, 63, 64], for positive pregnancy experiences.

Mothers referred for delivery at the KCMC zonal referral hospital were more likely to deliver preterm compared to those who had self-referred (normal clinic attendance). Similar findings has been reported elsewhere [43, 62], where women referred for delivery are more likely to have more pregnancy-related complications such as pre-eclampsia, which increases the risk of preterm birth. Close clinical follow-up is recommended to this group of women during prenatal care to minimize pregnancy-related complications, such as preterm birth and associated consequences. Mothers with primary education compared to higher (college/university) education level had significantly higher odds of delivering moderately to late, but not very/extremely preterm. These findings were consistent to a meta-analysis of 12 European Cohorts, where poor health at birth was higher among babies born from mothers with low education levels [65]. Low socio-economic status, including low education level is reported to affect pregnancy outcomes and complications [60, 66]. Policies and programs to improve maternal and child care in Tanzania should address health inequalities and prioritize the marginalized groups taking a multi-sectoral approach.

Furthermore, male children were more likely to be delivered preterm compared to females. This might be associated with shorter gestational duration for male compared to female fetuses [67]. A study in the UK found no significant relationship between fetal gender and the risk of preterm birth among women at high risk of delivering preterm (ie, with a history of miscarriage, preterm birth or cervical surgery) [18]. We also found that primiparous women were less likely to deliver preterm compared to multiparous. Findings from a meta-analysis using data from cohort studies in LMIC indicated that nulliparous, aged <18 years and parity ≥3 aged ≥35 years women were more likely to experience adverse neonatal outcomes, including preterm birth [63]. Other studies found no significant association between parity and the risk of preterm birth [5, 12, 32, 68]. Despite that, interventions to improve maternal and child care should be delivered through out the course of woman's reproductive period.

Among the factors associated with the rise in trends of preterm birth is the iatrogenic early delivery (i.e. following labour induction and/or caesarean delivery) carried out for fetal or maternal indications [69]. In this study, women who delivered moderately to late preterm were more likely to deliver through caesarean section (CS). It is possible that these women had other obstetric complications such as a previous CS, severe pre-eclampsia/eclampsia, placenta praevia, preterm premature rupture of membranes, and high birthweight that contributed highly to CS delivery and hence preterm birth [70, 71]. The odds of delivering both moderately to late and very/extremely preterm was high among mothers with pre-eclampsia/eclampsia, experienced placenta previa, and abruption placenta, as also reported elsewhere [5, 21, 60]. The effect of placenta previa on delivering very/extremely preterm were almost twice compared to the moderately to late preterm birth category. These conditions are both the risk factors as well as common indications for preterm birth [48, 60]. PROM increases the risk of preterm birth [9, 21, 48, 72], which is consistent to the findings in this study. Previous studies have shown that PROM is among the common indications of spontaneous preterm birth [21, 22, 72].

LBW was associated with eight-fold higher odds of moderately to late preterm ([32,37) weeks of gestation) and nearly 40 times higher odds of very/extremely preterm (<32 weeks of gestation). In fact, the proportions of moderately to late and very/extremely preterm birth were significantly higher among deliveries born with LBW than in the normal birth weight deliveries (37.1% and 14.3%, vs 6.4% and 0.4%, respectively) (results before imputation). Our findings agree with a previous case-control study in northern Tanzania, where LBW was associated with over 34-folds risk of preterm delivery [5]. The observed increase in preterm birth due to LBW could be attributed to two factors; the fact that preterm birth is also a risk factor for LBW (low birth weight but appropriate for gestation age) and intrauterine growth retardation or small for gestational age. Literature shows that extremely preterm babies are more likely to be born with LBW, while newborns small for gestational age are at a higher risk of experiencing morbidity and mortality [73, 74]. In this study, 81.2% (688/847) of very/extreme preterm newborns were born with both LBW and preterm compared to 41.6% (1779/4279) among moderately to late preterm (results before imputation). On the other hand, babies born preterm are at an increased risk of being born with LBW [44] and experiencing perinatal and neonatal morbidity and mortality [20, 43]. Care for the LBW and preterm babies is a critical intervention for improving child survival. Special attention should be given to babies born with LBW at <32 weeks of gestation.

According to the WHO recommendations, antenatal care visit remains to be a critical entry point where high-risk pregnancies can be identified and managed [24, 72, 75]. We found that women with inadequate (<4) ANC visits are more likely to deliver moderately to late and very/extremely preterm. Similar findings were also reported in other studies [5, 6, 48, 76]. However, these studies estimated the association between the number of ANC visits in the overall preterm birth categories (<37 weeks of gestation) compared to our study that showed different risk patters in two sub-categories of preterm birth (<32 and [32,37) weeks of gestation). In Tanzania, over half (51%) of pregnant women had at least four ANC visits during their last pregnancy [61]. Considering the current WHO recommendations of eight or more visits [75], different strategies are needed to promote health care seeking behaviors for pregnant women, and provision of quality ANC services at all levels of care. The timing and number of ANC visits is as important as the content and quality of care [77].

In this study, we applied the multinomial regression models with two categories of preterm birth (<32 and [32,37) weeks of gestation) due to rarity of cases in the <28 gestational weeks category. Eventually, the collapsed categories increased statistical power. Nevertheless, it is also possible that there may be under-reporting of extreme premature deliveries in the KCMC

Medical birth registry. Despite the low accuracy of gestational age estimation based on the date of last menstrual period [9, 10, 60], it remains the widely used method in resource-limited settings like Tanzania. Even where ultrasound is available, this method "requires skilled technicians, equipment and for maximum accuracy, first-trimester antenatal clinic attendance" [9], which is still a challenge in Tanzania [61]. There are alternative gestational age estimation methods, such as a combination of ultrasound and LMP [9, 10, 60], but the question remains on the feasibility and applicability of these options in resource-limited settings.

Another limitation of this study is that it was hospital-based, utilizing the KCMC Medical Birth Registry data from the KCMC zonal referral hospital in northern Tanzania, hence suffers from referral bias. Nearly a quarter of all women were referred for delivery during the study period. This may affect the generalization of the results. Nevertheless, this is the only birth-registry in the country (and potentially one of the few in SSA) providing critical information for pregnancy monitoring, administrative, and research purposes. Such registries allows for routine and inter-generational linkage and analysis of mother-child records. The KCMC hospital and its partners should promote routine data quality checks, resolve data quality and reporting challenges to ensure a sustainable operation of the birth registry, for current and future use.

## Conclusion

The findings from this study support other studies showing improved precision of parameter estimates after imputation of missing values and the rising trends of preterm birth rates. The multinomial regression models allowed for the simultaneous assessment of predictors of different preterm birth categories as opposed to binary regression analysis. Policy decisions should intensify efforts on improved maternal and child care throughout the course of pregnancy and childbirth, towards prevention of preterm birth. Interventions to increase the uptake and quality of ANC services should also be strengthened in Tanzania at all levels of care, where several interventions can easily be delivered to pregnant women [75], especially those at high-risk of experiencing adverse pregnancy outcomes. The number of ANC visits is as important as the content of care [77].

## Acknowledgments

We would like to acknowledge the midwives who participated in data collection and all women and children whose information enabled the availability of data used in this study. The authors also thank the staff at the Birth Registry for capturing these data in the electronic system. We also appreciate the Centre for International Health at the University of Bergen in Norway and the Department of Obstetrics and Gynecology of the KCMC hospital in Tanzania for establishing the KCMC medical birth registry, which facilitated the availability of data to conduct this study.

## Author Contributions

**Conceptualization:** Innocent B. Mboya, Michael J. Mahande, Joseph Obure, Henry G. Mwambi.

**Data curation:** Innocent B. Mboya, Michael J. Mahande.

**Formal analysis:** Innocent B. Mboya, Henry G. Mwambi.

**Investigation:** Innocent B. Mboya, Michael J. Mahande, Joseph Obure, Henry G. Mwambi.

**Methodology:** Innocent B. Mboya, Michael J. Mahande, Henry G. Mwambi.

**Project administration:** Innocent B. Mboya, Michael J. Mahande, Joseph Obure, Henry G. Mwambi.

**Supervision:** Michael J. Mahande, Joseph Obure, Henry G. Mwambi.

**Validation:** Innocent B. Mboya, Michael J. Mahande, Joseph Obure, Henry G. Mwambi.

**Visualization:** Innocent B. Mboya, Michael J. Mahande, Joseph Obure.

**Writing – original draft:** Innocent B. Mboya.

**Writing – review & editing:** Innocent B. Mboya, Michael J. Mahande, Joseph Obure, Henry G. Mwambi.

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
