## [Decision Letter · Decision Letter 0]

9 Feb 2021

PONE-D-20-37595

Predictors of singleton preterm birth using multinomial regression models accounting for missing data: a birth registry-based cohort study in northern Tanzania

PLOS ONE

Dear Dr. Mboya,

Thank you for submitting your manuscript to PLOS ONE. After careful consideration, we feel that it has merit but does not fully meet PLOS ONE’s publication criteria as it currently stands. Therefore, we invite you to submit a revised version of the manuscript that addresses the points raised during the review process.

This is a well written manuscript. All of the reviewers have suggested revisions which would help to improve the manuscript. I won't add to those, other than asking the authors to ensure that the manuscript is consistent with STROBE guidelines. Of the reviewers' comments, please pay particular attention to the "major" comments from reviewer 2 who was specifically recruited because of their expertise in statistical analyses - which are key for this manuscript. I look forward to seeing a revised manuscript.

We look forward to receiving your revised manuscript.

Kind regards,

Clive J Petry, PhD

Academic Editor

PLOS ONE

2. Please include additional information regarding the survey or questionnaire used in the study and ensure that you have provided sufficient details that others could replicate the analyses. For instance, if you developed a questionnaire as part of this study and it is not under a copyright more restrictive than CC-BY, please include a copy, in both the original language and English, as Supporting Information. If the questionnaire is published, please provide a citation to the questionnaire and/or original publication associated with the questionnaire.

3. Please include the date(s) on which you accessed the databases or records to obtain the retrospective data used in your study.

Reviewers' comments:

Reviewer's Responses to Questions

**Comments to the Author**

1. Is the manuscript technically sound, and do the data support the conclusions?

Reviewer #1: Yes

Reviewer #2: Yes

Reviewer #3: Yes

2. Has the statistical analysis been performed appropriately and rigorously? 

Reviewer #1: Yes

Reviewer #2: Yes

Reviewer #3: Yes

3. Have the authors made all data underlying the findings in their manuscript fully available?

Reviewer #1: Yes

Reviewer #2: Yes

Reviewer #3: Yes

4. Is the manuscript presented in an intelligible fashion and written in standard English?

Reviewer #1: Yes

Reviewer #2: Yes

Reviewer #3: Yes

5. Review Comments to the Author

Reviewer #1: Thank you for inviting me to review this paper which I enjoyed reading. The manuscript is well written, the methods are clear and the statistical analysis is appropriate. I appreciate that the authors have used a multinomial regression analysis strategy, and they have nicely explained the added-value of this approach when studying preterm birth. I recommend the paper for publication, and have only minor comments to make.

Reading linearly through the text:

In the methods section, the authors should clarify that the variable "delivered LBW" relates to the current delivery and not to "previous LBW delivery" as this is another well known maternal risk factor for preterm birth (PBT). Likewise for the perinatal status variable, do you mean stillbirth? and are these the babies at 28 weeks?

If so, I would add in the discussion that the authors acknowledge the etiologies and delivery practices for stillbirths vs. live births differ which might impact the results, however these represented a very small fraction of the births in the study.

Also in methods, you could explain if/why you have chosen to look at HIV mothers separately. In the results, I was a bit surprised that infections did not seem not associated with preterm birth.

In the discussion, the association between LBW and PTB can be explored more in depth as LBW can be both a predictor and an outcome of PTB. This is worth mentioning especially in the context of the multinomial analysis. For example, the odds of LBW in very preterm neonates are high and this is expected as many of these babies might be under 2500g due to (normal) intrauterine fetal growth velocity, as well as issues of being small for gestational age/fetal growth retardation.

Other minor edits :

the PPH acronym in Table 2 needs to be provided in full in the text, I didn't see it..

in the abstract, there is mention of "November 30?"

line 216 p. 7: A"s suggested by [35] " add author name instead of the reference number only.

In the methods section, the description of the imputation methodology could be shortened a bit as long as the appropriate reference is given and standard methods were used.

Reviewer #2: Summary

The authors use a multinomial regression model to predict singleton preterm birth in northern Tanzania. They also account for the presence of missing data.

My opinion

The article is interesting and well-written, and it deserves to be published after a revision which will take into account the following major/minor comments.

Major comments

• Multinomial Logistic Regression is a simple extension of the classical binomial logistic regression model to be used when the response variable has more than two nominal (unordered) categories. When the response categories are ordered, as for the response variable considered by the authors (preterm birth), one could always run a multinomial regression model. However, the disadvantage is that information about the ordering is throwing away. Instead, an ordinal logistic regression model preserves that information, although it is slightly more involved. The authors should use such an ad-hoc model or, at least, they should justify (trying to convince the reader) why they preferred using a multinomial logistic regression model.

• Several criteria have been proposed in the literature to compare model performance. In this regard, the authors should justify the use of the AIC.

• As any other statistical model, multinomial regression has some underlying assumptions too. These assumptions should be checked by the authors.

Minor comments

• Page 1 (Methods): Improve the notation “32-<37”. Perhaps, the authors could use [32,37).

• Page 1 (Results): Substitute “very/ extremely” with “very/extremely”.

• Line 110: Substitute “widow/ divorced” with “widow/divorced”.

• Line 110: Substitute “pre-eclampsia/ eclampsia” with “pre-eclampsia/eclampsia”. There are several typos of this kind.

• Lines 188-216: Instead of declaring the STATA commands used in the text, I suggest postponing, either in appendix or in a supplementary file, the whole STATA code to replicate the analysis.

• Equation (6): Revise $\\theta M$ in the first round brackets.

Reviewer #3: General remarks:

Preterm birth is a leading cause of neonatal mortality and a significant contributor to short and long- term morbidity. As the authors state the trends of preterm birth have increased in Tanzania during the years of the study and Tanzania has one of the highest rates of preterm birth in the world.

Thus, it is of great importance to improve the understanding of risks contributing to preterm birth.

The paper is well written and the findings are in agreement with previous studies on preterm birth. The discussion and conclusion are sound. The authors have used another statistical approach than some other papers in the field, but otherwise the paper does not contribute with new knowledge, but rather confirm previous knowledge.

Minor remarks:

LBW is categorized as birth weight below 2500g in the paper. However, LBW should be categorized as low birthweight according to gestational age. Birth weight of 2500g is not LBW if you are born extremely/very preterm.

If I understood it correctly the women were interviewed after giving birth? The data was not prospectively collected? Please clarify this and if that is correct I think that should be mentioned and discussed further in the limitation section of the discussion.

I would also like the authors to elaborate on how this paper contributes to previous knowledge in the field.

6. PLOS authors have the option to publish the peer review history of their article (what does this mean?). If published, this will include your full peer review and any attached files.

Reviewer #1: **Yes: **Marie Delnord

Reviewer #2: No

Reviewer #3: No

---

## [Author Response · Author response to Decision Letter 0]

17 Feb 2021

Reviewer reports:

Reviewer #1: 

Thank you for inviting me to review this paper which I enjoyed reading. The manuscript is well written, the methods are clear and the statistical analysis is appropriate. I appreciate that the authors have used a multinomial regression analysis strategy, and they have nicely explained the added-value of this approach when studying preterm birth. I recommend the paper for publication, and have only minor comments to make.

Response: We thank the reviewer for the very positive comment. We hope the manuscript can be accepted for publication after addressing the reviewer and editorial comments. 

Reading linearly through the text:

In the methods section, the authors should clarify that the variable "delivered LBW" relates to the current delivery and not to "previous LBW delivery" as this is another well-known maternal risk factor for preterm birth (PBT). Likewise, for the perinatal status variable, do you mean stillbirth? and are these the babies at 28 weeks?

Response: We thank the reviewer for this comment. Indeed, history of LBW and other adverse pregnancy outcomes increases the risk of recurrence and for other adverse outcomes. Some mothers in the KCMC medical birth registry had more than one child, hence their birth records available and all used in the analysis. It is possible that two children from the same mother would be born with LBW. We used robust variance estimator to account for clustering of these deliveries within a mother. Regarding perinatal status, this comprises stillbirths (pregnancy loss that occurs after seven months of gestation) and early neonatal death (death of live births within the first seven days of life). We have provided additional information with citations in page 5, line 125-126.

If so, I would add in the discussion that the authors acknowledge the etiologies and delivery practices for stillbirths vs. live births differ which might impact the results, however these represented a very small fraction of the births in the study.

Response: We thank the reviewer for this comment. Although the focus of this paper is on predicting preterm births, this outcome may also increase the risk of other adverse outcomes/events. We have provided the information about the increased risk of LBW and perinatal death in the discussion section, page 15, line 478-490.

Also in methods, you could explain if/why you have chosen to look at HIV mothers separately. In the results, I was a bit surprised that infections did not seem not associated with preterm birth.

Response: We would like to clarify that for this analysis, we did not separately look or analyze HIV mothers. Except in the descriptive analysis (Table 2), HIV status was not a significant predictor of preterm birth in this study as reported in other studies. Furthermore, the unobserved significant effect of infections might have been influenced by a very small percent of mothers (1.7% (765) of 44117) with infections. Of these 765, only 66 (8.6%) were born at 32-<37 weeks, 8 (1%) at 28-<32 weeks, and 2 (0.3%) at <28 weeks of gestation. These could be a reason for the unobserved effect of infections. We would also like to emphasize that the KCMC is a zona referral hospital in northern Tanzania, where the CEmOC services are available, for which such conditions are often well-managed. 

In the discussion, the association between LBW and PTB can be explored more in depth as LBW can be both a predictor and an outcome of PTB. This is worth mentioning especially in the context of the multinomial analysis. For example, the odds of LBW in very preterm neonates are high and this is expected as many of these babies might be under 2500g due to (normal) intrauterine fetal growth velocity, as well as issues of being small for gestational age/fetal growth retardation.

Response: We acknowledge the reviewer comment. We have revised the discussion section to reflect these comments. Changes are in the discussion, page 15, line 478-490. 

Other minor edits:

The PPH acronym in Table 2 needs to be provided in full in the text, I didn't see it.

Response: We acknowledge the reviewer comment. The PPH acronym has been provided in the study variables and variable definitions section, page 5, line 123. 

In the abstract, there is mention of "November 30?"

Response: We thank the reviewer for this comment. However, we could not find “November 30” mentioned in the abstract. 

line 216 p. 7: A"s suggested by [35] " add author name instead of the reference number only.

Response: We acknowledge the reviewer comment. Author name has been added in the statistical and computational analysis section, page 7, line 226. 

In the methods section, the description of the imputation methodology could be shortened a bit as long as the appropriate reference is given and standard methods were used.

Response: We acknowledge the reviewer comment. However, we prefer that the description of FCS algorithm be left in the paper for those who are the statistical inclined readers to benefit.

Reviewer #2: Summary

The authors use a multinomial regression model to predict singleton preterm birth in northern Tanzania. They also account for the presence of missing data.

My opinion

The article is interesting and well-written, and it deserves to be published after a revision which will take into account the following major/minor comments.

Response: We thank the reviewer for this comment. We hope the manuscript can now be accepted for publication following the revisions and responses to reviewer and editorial comments. 

Major comments

• Multinomial Logistic Regression is a simple extension of the classical binomial logistic regression model to be used when the response variable has more than two nominal (unordered) categories. When the response categories are ordered, as for the response variable considered by the authors (preterm birth), one could always run a multinomial regression model. However, the disadvantage is that information about the ordering is throwing away. Instead, an ordinal logistic regression model preserves that information, although it is slightly more involved. The authors should use such an ad-hoc model or, at least, they should justify (trying to convince the reader) why they preferred using a multinomial logistic regression model.

Response: We acknowledge this valid and very relevant comment by the editor. Indeed, by performing multinomial regression analysis the inherent ordering of preterm birth categories is lost. Although we also performed this analysis even before deciding to use multinomial regression models, we had the following concerns;

− Before imputation of missing values, there were a couple of variables that did not satisfy the proportional odds assumption, hence the OLOGIT model could not be used. The alternative model was the generalized ordered logit models (GOLOGIT). However, with this models, we encountered a non-convergence problem, especially with four preterm birth categories and appropriate interpretation of results as described below.

− Secondly, the order of gestational age categories is <28 weeks (extremely preterm), 28-32 weeks (very preterm), 32-<37 weeks (moderate to late preterm), and 37+ weeks (term). Assume the levels of this outcome are coded 0 to 3 (with 0 being term birth), the first panel of coefficients will be interpreted as; 0 vs. 1+2+3, then 0+1 vs 2+3 etc. This will imply modelling the probability of delivering at a normal gestational age (category 0) compared to preterm (categories 1-3), probability of delivering term + very preterm vs other preterm categories etc. Such interpretation we though would be somehow misleading given the nature of this outcome and may not be appealing to clinicians or public health practitioners. The same concern will be even if we reverse code the preterm birth categories from extremely preterm (0) to term (3). 

− Finally, the choice between these regression models often depends on the research question one would like to address. For us, we chose to use multinomial regression models to provide evidence for the predictors of preterm birth across different preterm birth categories (a multinomial variable) vs a binary variable where gestational age is coded as normal birth (37+ weeks of gestation) and preterm birth (<37 weeks of gestation) as with the majority of previous literature, including in Tanzania Setting. The multinomial regression model (MLOGIT) is one of the best alternatives to the OLOGIT model in this respect. 

• Several criteria have been proposed in the literature to compare model performance. In this regard, the authors should justify the use of the AIC.

Response: We acknowledge the reviewer comment. The AIC was used for model comparison whereby, the MLOGIT model had an AIC of 26546.73 compared to 26673 for the OLOGIT model. AIC is often used to compare the non-nested models. The likelihood ratio chi-square test was also used to compare nested models. The decision to use and report results from the MLOGIT model as reported in our manuscript was reached after carrying out several sensitivity analyses. 

• As any other statistical model, multinomial regression has some underlying assumptions too. These assumptions should be checked by the authors.

Response: We acknowledge the reviewer comment. Data analysis included testing the underlying assumption for the multinomial regression model. We would like to indicate it here that, one critical issue observed here is that the p-value from the multinomial goodness-of-fit chi-square test was overwhelmingly statistically significant (p<0.001). This suggested that the mlogit model was not a better fit. However, we carried out several sensitivity analyses such as performing the ordered logistic regression, multinomial probit regression, and even binary logistic regression analysis and got the same results on model fit. We further attempted to manually enter and remove variables from the final model to assess how the models will perform with no significant improvement. One possible explanation could be the sensitivity of the Chi-square goodness-of-fit test for large sample size which tends to provide statistically significant results. 

Minor comments

• Page 1 (Methods): Improve the notation “32-<37”. Perhaps, the authors could use [32,37).

Response: We thank the reviewer for this comment. We have made the recommended changes throughout the manuscript. 

• Page 1 (Results): Substitute “very/ extremely” with “very/extremely”.

Response: We thank the reviewer for this comment. We have made the recommended changes throughout the manuscript. 

• Line 110: Substitute “widow/ divorced” with “widow/divorced”.

Response: We thank the reviewer for this comment. We have made the recommended changes. 

• Line 110: Substitute “pre-eclampsia/ eclampsia” with “pre-eclampsia/eclampsia”. There are several typos of this kind.

Response: We thank the reviewer for this comment. We have made the recommended changes throughout the manuscript. 

• Lines 188-216: Instead of declaring the STATA commands used in the text, I suggest postponing, either in appendix or in a supplementary file, the whole STATA code to replicate the analysis.

Response: We thank the reviewer for the comment. We have removed all STATA commands from the manuscript and provided proper citation. We hope the revisions will address the reviewer concerns. 

• Equation (6): Revise $\\theta M$ in the first round brackets.

Response: We acknowledge the reviewer comment. Equation (.6) has been revised accordingly. 

Reviewer #3: 

General remarks:

Preterm birth is a leading cause of neonatal mortality and a significant contributor to short and long- term morbidity. As the authors state the trends of preterm birth have increased in Tanzania during the years of the study and Tanzania has one of the highest rates of preterm birth in the world. Thus, it is of great importance to improve the understanding of risks contributing to preterm birth.

The paper is well written and the findings are in agreement with previous studies on preterm birth. The discussion and conclusion are sound. The authors have used another statistical approach than some other papers in the field, but otherwise the paper does not contribute with new knowledge, but rather confirm previous knowledge.

Response: We acknowledge the reviewer’s relevant comments. We hope the manuscript can now be accepted for publication. 

Minor remarks:

LBW is categorized as birth weight below 2500g in the paper. However, LBW should be categorized as low birthweight according to gestational age. Birth weight of 2500g is not LBW if you are born extremely/very preterm.

Response: We acknowledge the reviewer comment. During our analysis, we had the same concern regarding the categorization of birth weight. We understand that it should be categorized according to gestational age. However, upon doing that, the analysis of LBW variable generated according to gestational age failed to converge even in the crude/un-adjusted analysis. For this reason, we decided to include LBW variable as it is in our analyses. For this reason, we defined LBW as an absolute infant birth weight of <2500g regardless of gestational age [1]. Changes are in the study variables section, page 5, line 126-127. We have also We have also revised the discussion on the association between LBW and preterm birth. Changes in Line 478-490. We understand that both LBW due to preterm and also, LBW due to small for dates are at increased risk of poor neonatal outcomes.

If I understood it correctly, the women were interviewed after giving birth? The data was not prospectively collected? Please clarify this and if that is correct I think that should be mentioned and discussed further in the limitation section of the discussion.

Response: Women were interviewed within 48 hours after delivery. The vast majority of them were interviewed within 24 hours, except those who experienced delivery complications. The KCMC Medical birth registry collects prospective data for all mothers and their subsequent deliveries in the hospital’s department of obstetrics and gynecology. We have provided some additional information on the data collection methods section, page 4, lime 90-92. 

I would also like the authors to elaborate on how this paper contributes to previous knowledge in the field.

Response: We thank the reviewer for this comment. Tanzania currently ranks the 10th country with the highest preterm birth rates in the world. Globally, preterm birth rates are on the rise. Studies are needed to understand current trends and potential risk factors for informed interventions. This study complements previous studies on this outcome by showing trends of preterm birth over a 15-years period to inform clinical, administrative and public health decisions. The study also determined the risk of preterm birth across two gestational age categories as opposed to performing binary analysis. For more robust parameter estimates, data analysis accounted for missing values as opposed to previous studies that performed complete case analysis. Generally, our findings on the predictors of preterm birth agrees with the previous literature. 

Reference

1. Cutland, C.L., Lackritz, E.M., Mallett-Moore, T., Bardají, A., Chandrasekaran, R., Lahariya, C., Nisar, M.I., Tapia, M.D., Pathirana, J., Kochhar, S. and Muñoz, F.M., 2017. Low birth weight: Case definition & guidelines for data collection, analysis, and presentation of maternal immunization safety data. Vaccine, 35(48Part A), p.6492.

---

## [Decision Letter · Decision Letter 1]

12 Mar 2021

PONE-D-20-37595R1

Predictors of singleton preterm birth using multinomial regression models accounting for missing data: a birth registry-based cohort study in northern Tanzania

PLOS ONE

Dear Dr. Mboya,

Thank you for submitting your manuscript to PLOS ONE. After careful consideration, we feel that it has merit but does not fully meet PLOS ONE’s publication criteria as it currently stands. Therefore, we invite you to submit a revised version of the manuscript that addresses the points raised during the review process.

The manuscript has been improved by the various revisions made. Reviewer 2 still feels that there are a few things that have not yet been completed adequately, so has suggested further (minor) revisions. I agree that they would help make the manuscript as good as possible. They should not be too onerous to complete. I look forward to seeing the final version of the manuscript. 

We look forward to receiving your revised manuscript.

Kind regards,

Clive J Petry, PhD

Academic Editor

PLOS ONE

Journal Requirements:

Reviewers' comments:

Reviewer's Responses to Questions

**Comments to the Author**

1. If the authors have adequately addressed your comments raised in a previous round of review and you feel that this manuscript is now acceptable for publication, you may indicate that here to bypass the “Comments to the Author” section, enter your conflict of interest statement in the “Confidential to Editor” section, and submit your "Accept" recommendation.

Reviewer #1: (No Response)

Reviewer #2: All comments have been addressed

2. Is the manuscript technically sound, and do the data support the conclusions?

Reviewer #1: Yes

Reviewer #2: Yes

3. Has the statistical analysis been performed appropriately and rigorously? 

Reviewer #1: Yes

Reviewer #2: Yes

4. Have the authors made all data underlying the findings in their manuscript fully available?

Reviewer #1: Yes

Reviewer #2: Yes

5. Is the manuscript presented in an intelligible fashion and written in standard English?

Reviewer #1: Yes

Reviewer #2: Yes

6. Review Comments to the Author

Reviewer #1: Thank you to the authors for properly addressing my suggestions in the revision. No further comments besides minor edits for the proofs:

- the authors should add “weeks GA” in the legend of Fig 2

- and fix this sentence: “Tanzania has also adopted these 45 strategies [26, 27] and is one of the five countries where WHO implements a clinical trial 46 on the immediate kangaroo mother care (KMC) for women at risk of preterm birth [2, 27].” The wording is not clear as immediate kangaroo care is used when the baby is born, so technically mothers are not “ at risk” of preterm birth, the outcome has happened already.

Reviewer #2: The authors did a good enough job in answering to my questions in the “Response to Reviewers” part of the pdf. However, they failed to include part of the arguments given to me into the paper. Therefore, I have still some minor comments for them.

1. I think that many of the arguments used by the authors to justify the use of multinomial logistic regression, over the ordinal logistic regression model, should be added to the paper. This could be useful for readers having my same initial concerns.

2. As for the use of the AIC for comparing non-nested models, there is some disagreement in the literature. This is mainly due to the fact that the original theory by Akaike works out for nested models only. The authors should cite some work available in the literature justifying the use of the AIC for comparing non-nested models.

3. The authors often answer to me by saying that they carried out several sensitivity analyses to decide about what including into the paper. I wonder if it is not appropriate to include the main results of all these analyses in the paper.

7. PLOS authors have the option to publish the peer review history of their article (what does this mean?). If published, this will include your full peer review and any attached files.

Reviewer #1: No

Reviewer #2: No

---

## [Author Response · Author response to Decision Letter 1]

15 Mar 2021

Reviewer #1: 

Thank you to the authors for properly addressing my suggestions in the revision. No further comments besides minor edits for the proofs:

- the authors should add “weeks GA” in the legend of Fig 2

Response: We thank the reviewer for this observation. We have included this information in the legend of Fig 2. 

- and fix this sentence: “Tanzania has also adopted these 45 strategies [26, 27] and is one of the five countries where WHO implements a clinical trial 46 on the immediate kangaroo mother care (KMC) for women at risk of preterm birth [2, 27].” The wording is not clear as immediate kangaroo care is used when the baby is born, so technically mothers are not “at risk” of preterm birth, the outcome has happened already.

Response: We acknowledge the reviewer comment. We have corrected this sentence as suggested. Changes are in the introduction section, page 3, line 47-48. 

Reviewer #2: 

The authors did a good enough job in answering to my questions in the “Response to Reviewers” part of the pdf. However, they failed to include part of the arguments given to me into the paper. Therefore, I have still some minor comments for them.

1. I think that many of the arguments used by the authors to justify the use of multinomial logistic regression, over the ordinal logistic regression model, should be added to the paper. This could be useful for readers having my same initial concerns.

Response: We agree with the reviewer. We have added this information in the statistical and computational analysis section, page 5, lines 149-165 to page 6, line 166-169. 

2. As for the use of the AIC for comparing non-nested models, there is some disagreement in the literature. This is mainly due to the fact that the original theory by Akaike works out for nested models only. The authors should cite some work available in the literature justifying the use of the AIC for comparing non-nested models.

Response: We acknowledge the reviewer comment. While we have provided a citation on the use of AIC to compare non-nested models, we still cannot ignore the existing debate on this subject. Because we fit several models, we used AIC to compare their performance. The application of several other information criteria in comparing model performance is an area of further investigation. 

3. The authors often answer to me by saying that they carried out several sensitivity analyses to decide about what including into the paper. I wonder if it is not appropriate to include the main results of all these analyses in the paper.

Response: We thank the reviewer for this comment. However, we chose not to include these results because the focus was to show the results from the multinomial logistic regression model. Also, results from other models, such as the ologit, and gologit models could not be presented due to issues on model performance, as previously indicated. We hope a brief description in the statistical and computational analysis section, page 5, lines 149-165 to page 6, line 166-169 will briefly highlight some of the issues observed in our analyses.

---

## [Editor Report · Decision Letter 2]

18 Mar 2021

Predictors of singleton preterm birth using multinomial regression models accounting for missing data: a birth registry-based cohort study in northern Tanzania

PONE-D-20-37595R2

Dear Dr. Mboya,

We’re pleased to inform you that your manuscript has been judged scientifically suitable for publication and will be formally accepted for publication once it meets all outstanding technical requirements.

Kind regards,

Clive J Petry, PhD

Academic Editor

PLOS ONE
---

## [Editor Report · Acceptance letter]

22 Mar 2021

PONE-D-20-37595R2 

Predictors of singleton preterm birth using multinomial regression models accounting for missing data: a birth registry-based cohort study in northern Tanzania 

Dear Dr. Mboya:

I'm pleased to inform you that your manuscript has been deemed suitable for publication in PLOS ONE. Congratulations! Your manuscript is now with our production department. 

Kind regards, 

on behalf of

Dr. Clive J Petry 

Academic Editor

PLOS ONE